# Liver Stiffness in Obese Hypothyroid Patients Taking Levothyroxine

**DOI:** 10.3390/medicina58070946

**Published:** 2022-07-18

**Authors:** Roberta Pujia, Elisa Mazza, Tiziana Montalcini, Franco Arturi, Antonio Brunetti, Antonio Aversa, Stefano Romeo, Maria Perticone, Angela Sciacqua, Arturo Pujia

**Affiliations:** 1Department of Medical and Surgical Science, University Magna Grecia of Catanzaro, 88100 Catanzaro, Italy; roberta.puj@gmail.com (R.P.); elisamazza@unicz.it (E.M.); arturi@unicz.it (F.A.); romeo@unicz.it (S.R.); mariaperticone@unicz.it (M.P.); sciacqua@unicz.it (A.S.); pujia@unicz.it (A.P.); 2Department of Clinical and Experimental Medicine, University Magna Grecia of Catanzaro, 88100 Catanzaro, Italy; aversa@unicz.it; 3Department of Health Science, University Magna Grecia of Catanzaro, 88100 Catanzaro, Italy; brunetti@unicz.it; 4Department of Molecular and Clinical Medicine, The University of Gothenburg, 40530 Gothenburg, Sweden

**Keywords:** thyroid, liver fibrosis, Levothyroxine, obesity, transient elastography, hypothyroidism

## Abstract

*Background and Objectives*: Thyroid dysfunction is associated with non-alcoholic fatty liver disease, but its role in the progression of liver damage in obese patients remains unclear. In addition, several case reports have suggested the existence of a levothyroxine-induced liver injury, which has been poorly investigated. Our aim was to verify whether a difference in the prevalence of liver fibrosis exists in a population of obese individuals taking Levothyroxine. *Materials and Methods*: We conducted a cross-sectional study on a population of 137 obese individuals, of which 49 were on replacement therapy with Levothyroxine. We excluded those who had hypertriglyceridemia and diabetes mellitus. All participants underwent a liver stiffness assessment by transient elastography as well as biochemical measurements. In subjects with liver fibrosis, other cause of liver fibrosis were ruled out. *Results*: Participants taking Levothyroxine had a higher prevalence of liver fibrosis than those not taking Levothyroxine (30.6% vs. 2.3%; *p* < 0.001), and these results were obtained after we made an adjustment for age (Exp(B) = 18.9; 95% CI = 4.1–87.4; *p* < 0.001). The liver stiffness value differed significantly between groups (6.0 ± 3.6 and 5.1 ± 1.2, *p* = 0.033). Of those subjects taking Levothyroxine, there were no significant differences in the dose of medication (1.21 ± 0.36 vs. 1.07 ± 0.42; *p* = 0.240) and treatment duration (13.7 ± 7.43 vs. 11.13 ± 6.23; *p* = 0.380) between those with and without liver fibrosis. *Conclusions*: We found, for the first time, a greater prevalence of liver fibrosis in obese individuals taking Levothyroxine than in those not taking this medication. This finding needs to be confirmed by longitudinal population studies as well as by cellular studies.

## 1. Introduction

Several studies have reported that non-alcoholic fatty liver disease (NAFLD) is associated with overt or subclinical hypothyroidism [1,2]. The plausibility of the link between liver steatosis and thyroid hormones is based on the role of the latter as regulators of lipid and carbohydrate metabolism, body weight, and hepatic insulin sensitivity [3,4,5,6,7,8,9]. Hypothyroidism is considered a risk factor for developing NAFLD [10]. In addition, the liver is important for thyroid function, since it is involved in thyroid hormone peripheral metabolism, oxidative deamination, biliary excretion, and the deiodination of thyroxine (T4) to triiodothyronine (T3) and to reverse T3 [7]. In addition, the liver produces thyroid hormone-binding proteins, such as pre-albumin, albumin, and thyroxine-binding globulin. However, the relationship between NAFLD and the thyroid function remains unclear [11,12,13].

Thyroid hormones also affect body weight by regulating the energy homeostasis, lipogenesis, and basal metabolic rate [14,15]. A high-normal serum thyroid stimulation hormone (TSH) is associated with a high body mass index (BMI) [16]. On the other hand, it has been demonstrated that adipose tissue influences survival and differentiation of thyrocytes [17]. Obesity is closely related with NAFLD due to ectopic fat accumulation in the liver and hepatic insulin resistance [18], and insulin resistance has been found in hypothyroidism [19]. Since thyroid hormones modulate the synthesis of collagen and the fibrogenic response in hepatic stellate cells [20], thyroid dysfunction may also be involved in the progression of NAFLD beyond simple steatosis [10,21]. Liver fibrosis is a repair response to chronic injury and represents the underlying pathogenic driver of carcinogenesis [22]. In this regard, hepatocellular carcinoma (HCC) tissues overexpress thyroid stimulating hormone receptors (TSHRs) [23]. At present, a thyroid hormone receptor beta agonist is being tested for the treatment of NASH by Kelly et al. [24]. Interestingly, few case reports have showed that the administration of exogenous thyroid hormones, such as Levothyroxine (L-T4), could cause liver injury with [25,26,27,28,29,30,31] with a mechanism that remains unclear. While the majority of studies have investigated the relation between obesity, NAFLD and thyroid dysfunction, to date there is a lack of data regarding liver fibrosis. It is unclear whether there is a high prevalence of liver fibrosis in individuals affected by thyroid dysfunction and what role autoimmunity and medications, such as L-T4, play in inducing liver damage. The aim of our study, which we believe is the first on this issue, was to assess the difference in the prevalence of liver fibrosis in a population of obese individuals taking or not taking L-T4.

## 2. Materials and Methods

In this cross-sectional study, the population was composed of adult obese outclinic patients regularly attending the Endocrinology Unit of the “Mater Domini” University Hospital in Catanzaro (Italy), between October 2018 and November 2019. After signing an informed consent, the patients were invited to be screened for the possible presence of liver disease by transient elastography (TE) at the Outpatient Clinic of the Clinical Nutrition Unit in the same University Hospital. A sample of 137 individuals were included in this study; they were aged between 35 and 80, and 49 were on replacement therapy for hypothyroidism with L-T4. We excluded those who had past and current alcohol abuse (>20 g of alcohol per day; 350 mL (12 oz) of beer, 120 mL (4 oz) of wine, and 45 mL (1.5 oz) of hard liquor each contain 10 g of alcohol), impaired liver function, presence of cholestatic liver disease, liver cirrhosis, pregnancy, nephrotic syndrome, chronic renal failure, cancer, and those who had incomplete data as ascertained from their clinical records. We excluded all those with FT3 and FT4 not in normal range, patients taking dietary supplements or psychotropic drugs, and those with current or past use of potential hepatotoxic drugs (e.g., methyldopa, tetracyclines, amiodarone, methotrexate, sodium valproate, prednisone, tamoxifen, metformin). We also excluded all those with serum triglycerides ≥ 250 mg/dL, type 1 and type 2 diabetes mellitus (T1DM, T2DM).

All participants underwent a brief interview to obtain information regarding their thyroid disease (causes and duration), current physical exercise, smoking habits, and medication (duration, dosage, and formulation). Each patient underwent anthropometric and laboratory assessments. To diagnose liver fibrosis, we focused on the stiffness value measured with TE. Blood pressure was determined at the time of the visit. The following criteria were used to define and exclude diabetes: fasting blood glucose ≥ 126 mg/dL or antidiabetic treatment [32]. The investigation conforms to the principles outlined in the Declaration of Helsinki.

### 2.1. Biochemical Evaluation

Serum glucose, total cholesterol, high density lipoprotein cholesterol, triglycerides, alanine transaminase, aspartate transaminase, gamma-glutamyl-transferase, total bilirubin, alkaline phosphatase, TSH, free triiodothyronine and free thyroxin were evaluated after fasting overnight. Low-density lipoprotein (LDL) cholesterol level was calculated with the Friedewald formula [33].

The subjects affected by liver fibrosis underwent the following diagnostic tests to rule out secondary forms of liver fibrosis: Anti-mitochondrial antibodies (AMAs), anti-liver kidney microsomal antibodies (Anti LKMs), anti-smooth muscle antibodies (ASMAs), anti-nuclear antibodies (ANAs), anti-centromere antibodies (ACAs), IgM and IgG antibodies to Cytomegalovirus, anti-hepatitis C virus antibodies, hepatitis B surface antigen, hepatitis B surface antibody, alpha-1-antitrypsin, and serum ceruloplasmin [34].

### 2.2. Anthropometric Measurements

Body weight was measured after a 12 h overnight fast. Body weight was measured on a calibrated digital scale (model Tanita BC-418MA) accurate to 0.1 kg, and standing height was measured with a wall-mounted stadiometer (TANITA, Middlesex, UK) [35]. BMI was calculated with the following equation: weight (kg)/height (m^2^). Obesity was defined by the presence of a body mass index (BMI) ≥ 30 kg/m^2^ [36]. Obesity classes were defined as I (BMI ≥ 30 < 35), II (BMI ≥ 35 < 40), III (BMI ≥ 40).

### 2.3. Liver Transient Elastography

TE quantifies liver steatosis by controlled attenuation parameter (CAP) assessment and measure liver stiffness (Fibroscan^®^; Echosense SASU, Paris, France) [37,38]. Both the CAP and stiffness score were obtained simultaneously as described in previous study (38). Since we enrolled only obese, we used the XL probe while we used M probe for CAP measurement. The tip of the probe transducer was placed on the skin between the rib bones at the level of the right lobe of the liver. All scans were performed by the same in-vestigator, with subjects fasting for at least 4 h. Liver stiffness was expressed by the median value (in kPa) of ten measurements performed between 25 and 65 mm depth. Only results with 10 valid shots and interquartile range (IQR)/median liver stiffness ratio < 30% were included. The cut-off value for defining the presence of fibrosis was liver stiffness > 7 kPa [37,39,40,41]. We assessed CAP score using only the M probe because the CAP algorithm is specific to this device. Ten successful measurements were performed on each patient, and only cases with ten successful acquisitions were taken into account for this study. The success rate was calculated as the number of successful measurements divided by the total number of measurements. The ratio of the IQR of liver stiffness to the median (IQR/MLSM) was calculated as an indicator of variability. The final CAP score (ranged from 100 to 400 decibels per meter (dB/m), was the median of individual measurements. The ratio of IQR in CAP values to the median (IQR/M CAP) was used as an indicator of variability for the final CAP. The diagnosis of NAFLD was based on a CAP > 246 dB/m. In order to identify each steatosis grade, three different cut-offs were used: CAP between 247 and 268 dB/m for the diagnosis of S1 grade, CAP between 269 and 280 dB/m for the diagnosis of S2 grade, and CAP > 281 dB/m for the diagnosis of S3 grade (severe) [42].

### 2.4. Statistical Analysis

Data were reported as mean ± SD. We categorized the population according with the use or not of L-T4. Then, a chi square test was performed to compare the prevalence and a Student’s *t*-test was performed to compare the means between the two groups. A logistic regression analysis was used to adjust liver fibrosis prevalence for age. In addition, stepwise multivariable linear regression analysis was used to test the association between liver stiffness, serving as the dependent variable, and all of the potentially confounding variables served as independent variables. These potential confounders were all those factors correlated with liver stiffness in the Pearson’s correlation with a *p* value < 0.1 (age, BMI, triglycerides, glucose, HDL), given that the continuous variables were normally distributed. Continuous data normality was analyzed through the Shapiro-Wilk test and Box plot method. We excluded the outliers from the analysis. Significant differences were assumed to be present at *p* < 0.05 (two-tailed). All analyses were performed using SPSS 20.0 for Windows (S. Wacker Drive, Chicago, IL, USA).

On the basis of the literature, the fibrosis prevalence was assumed to be 5%. A sample of at least 35 subjects in each group would be sufficient to detect a difference of 25% in prevalence with an alpha of 0.05 and a power of 80%.

## 3. Results

The mean age of the population was 55 ± 10 years. Mean BMI was 34.5 ± 4 kg/m^2^. Two diabetic patients and one subject with hypertriglyceridemia were excluded from the study. None of the subjects affected by fibrosis had a viral or autoimmune hepatitis. The mean CAP score was 274.03 ± 50 dB/m and the mean liver stiffness was 5.46 ± 2 kPa. A total of 112 (81%) subjects were female. In this population, 58% had liver steatosis and 12.4% had liver fibrosis. In addition, of those on L-T4 therapy, 55.1% had Chronic Autoimmune Thyroiditis, 20.5% had undergone surgery due to multinodular goiter, 12.2% had undergone surgery due to thyroid cancer, and 12.2% were affected by unspecified thyroid disease. There was no difference in fibrosis among the above indicated disease (data not shown). As expected, the BMI correlated with the liver stiffness (r = 0.22, *p* = 0.009; result not showed). The mean duration of L-T4 therapy was 12.7 ± 7 years. The mean dose was 1.15 ± 0.38 µg/kg/day. Table 1 shows the participants’ clinical characteristics according to L-T4 use. There were no differences between the two groups regarding CAP score, nor were there any differences in hepatic steatosis and gender prevalence (Table 1). However, liver stiffness was significantly different between groups (Table 1).

Liver fibrosis was more prevalent in the group on L-T4 treatment (found in 15 participants) than in the group not taking treatment (found in 2 participants) (Table 1), and this difference remained unchanged after adjustment for age (exp(B) = 18.9; 95% CI = 4.1–87.4; *p* < 0.001).

The laboratory parameters did not differ in the population according to the L-T4 treatment (Table 2). Among those on L-T4 treatment, there were no significant differences in the dose (1.21 ± 0.36 vs. 1.07 ± 0.42, *p* = 0.240), duration (13.7 ± 7.43 vs. 11.13 ± 6.23, *p* = 0.380), the formulation packages, and the causes of hypothyroidism, between those with and without fibrosis.

To corroborate the role of obesity, we analyzed 22 subjects recruited in previous steatosis trial. All were in Levo-T4 treatment with similar age (54.1), female prevalence (85%), and CAP score (285 dB/m) to the study group and with a BMI < 30 kg/m^2^. Unless they were all affected by steatosis (since it was an inclusion criterion), nobody had fibrosis (data not shown).

BMI, age, glycaemia, triglycerides did not influence the liver stiffness value (Table 3). Figure 1 shows the prevalence of obesity grade of the subjects, respectively, according to the L-T4 treatment.

## 4. Discussion

To the best of our knowledge, in a population of hypothyroid obese individuals taking L-T4, this is the first study to demonstrate a higher prevalence of liver stiffness as well as a higher stiffness value, compared to matched obese individuals not hypothyroid and not taking this medication (Table 1). This finding was not influenced by any other variables that were correlated with liver stiffness at univariate analysis (e.g., BMI, age, triglycerides, glycemia; see Table 3), nor by L-T4 dosage or formulation used. Since diabetes and dyslipidemia may drive the progression of steatosis toward fibrosis, and to better dissect the role of L-T4 treated hypothyroidism on the liver damage, we excluded individuals with diabetes and dyslipidemia. These two conditions are, in fact, characterized by elevated circulating free fatty acid levels and result in increased transport of free fatty acids to the liver predisposing to liver steatosis and fibrosis [43]. There was no difference regarding obesity classes between the two groups (Figure 1). Furthermore, there was no difference in the prevalence of hepatic steatosis between groups, as obese subjects have a high prevalence of NAFLD [44]. In our obese participants not taking L-T4, the prevalence of fibrosis was similar to that reported by Huh et al. in a population of metabolically healthy obese (2.3 vs. 3.8%) [45]. Indeed, the prevalence of fibrosis in our subjects taking L-T4 would remain significantly higher than that found by Huh et al. in metabolically unhealthy obese subjects (30.6 vs. 8.7%) [45]. Since some patients under treatment had a TSH higher than 4.0 µU/mL, and according some authors the goal of L-T4 should be a TSH less than 4.0 [46], we repeated the statistical analysis excluding subjects with TSH more than 4.0 µU/mL and the significance of the differences observed improved (*p* < 0.018 for stiffness; *p* < 0.0001 for fibrosis prevalence, data not shown). Despite previous reports suggesting the detrimental effects of L-T4 on the promotion of liver damage [25,26,27,28,29], its definitive role on liver fibrosis has been scarcely studied in humans. Liver damage during L-T4 therapy seems more prevalent in some populations (Asiatic) [28]. There have also been a few cellular investigations into the association between thyroid hormones and liver fibrosis [23,26]. Our study was not designed to explore the mechanisms underlying the pathogenic role of L-T4 on liver fibrosis, but the results suggest that there may be an association, thus generating hypotheses for future research. Our findings raise several important issues regarding the independent influence of L-T 4 on the liver. Whether the difference in liver stiffness is attributable to the chronic exposure to low concentrations of endogenous thyroid hormones or to the long-term effects of this medication remains to be clarified. Several hypotheses are currently being considered in the pathogenesis of liver fibrosis. The first considers that liver fibrosis is caused by self-immunity phenomena. However, in our study we excluded all individuals with autoimmune hepatitis. The second hypothesis is that liver fibrosis is part of the clinical picture of hypothyroidism. Indeed, some authors hypothesized that the injured liver may be due to a state of intrahepatic hypothyroidism. During liver repair, it has been reported a reduction of hepatocyte expression of deiodinase 1, an enzyme that converts thyroxine in its biologically active form, and an increase of stromal expression of deiodinase 3, that converts thyroxine into inactive hormone. As consequence, an accumulation of biologically inert thyroid hormone at the expense of biologically active one occurred, potentially altering hepatic differentiation as well as systemic thyroid hormones homeostasis, thereby contributing to negative outcomes of fibrosing liver injury [7]. This suggests that L-T4 only normalizes TSH but does not block the consequences of the disease (which may be a new finding never previously described). The third hypothesis includes the potential toxic effect of L-T4 on the liver cellular components. However, our study did not focus on these mechanisms. Our study is limited by its cross-sectional design. It is thus impossible to establish the time of onset of liver fibrosis first occurred (before or after L-T4 treatment), nor is it possible to infer causality. Whether the difference in liver stiffness is attributable to the long-term effects of this medication or chronic exposure to altered concentrations of endogenous thyroid hormones thus still needs to be clarified. The sample size is small. Another limitation is the lack of magnetic resonance use to confirm the findings of fibrosis. We recognize that in some studies [47] a discrepancy was reported between MRI and TE, but a recent review [48] confirms the validity of TE in the evaluation of fibrosis, especially of advanced one. We therefore replicated the test with a greater stiffness cut-off (>7.9 kPa) and the significant difference remains unchanged. Furthermore, any error of assessment would reflect on both groups. Moreover, American Diabetes Association guidelines suggest that elastography may be used to assess risk of fibrosis [49]. Longitudinal studies are thus needed to clarify the cause-effect link.

## 5. Conclusions

In conclusion, we found, for the first time, in a population of euthyroid obese individuals taking L-T4, a higher prevalence of liver stiffness compared to individuals not taking this medication. If confirmed by subsequent human longitudinal as well as cellular studies, these results suggest that modification of clinical practice is necessary and urgent. For example, instrumental tests could be performed to assess the presence of liver fibrosis in patients affected by hypothyroidism at the time of diagnosis, monitoring hepatic function in patients taking L-T4, or examining new perspectives for the treatment of hypothyroidism.

## Figures and Tables

**Figure 1 medicina-58-00946-f001:**
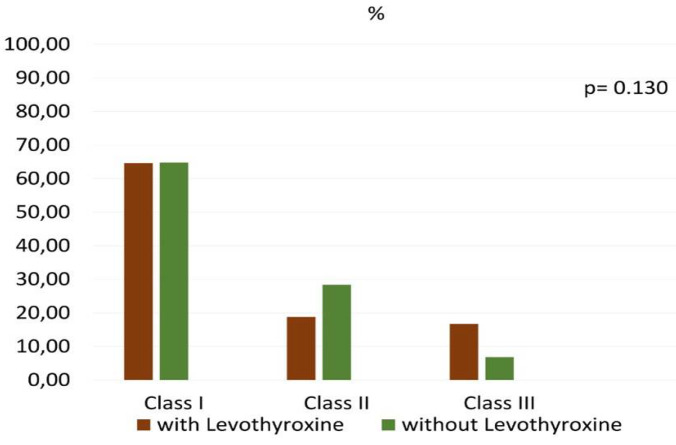
Prevalence of the different obesity classes according to Levothyroxine use.

**Table 1 medicina-58-00946-t001:** Mean +/− SD Participants’ clinical characteristics according to L-T4 use.

	Without L-T4 *n* = 88	With L-T4 *n* = 49	*p* Value
Age (years)	53.6 ± 9.6	58.1 ± 10.3	0.011
BMI (Kg/m^2^)	34.3 ± 3.5	34.8 ± 4.5	0.440
CAP score (dB/m)	270 ± 50	282 ± 50	0.258
Liver Stiffness (kPa)	5.1 ± 1.2	6.0 ± 3.6	0.033
Prevalence			
Liver fibrosis (%)	2.3	30.6	<0.001
Female (%)	77.3	87.8	0.174
Liver steatosis (%)	71.8 ^a^	75.0 ^b^	0.463
Hyperlipidemia (%)	38.6	28.6	0.150

Abbreviations: BMI, body mass index; CAP, Controlled Attenuation Parameter; L-T4, Levothyroxine; *p* value, probability value. ^a^ *n* = 78 because ten missing values. ^b^ *n* = 32 because seventeen missing values.

**Table 2 medicina-58-00946-t002:** Mean +/− SD Participants’ laboratory characteristics according to L-T4 use.

	Without L-T4 *n* = 88	With L-T4 *n* = 49	*p* Value
Glucose (mg/dL)	93.11 ± 10.13	96.63 ± 9.81	0.085
Total Cholesterol (mg/dL)	201.80 ± 35.13	190.05 ± 33.59	0.074
Triglycerides (mg/dL)	124.80 ± 48.02	132.88 ± 40.01	0.350
HDL Cholesterol (mg/dL)	53.36 ± 13.98	48.45 ± 11.43	0.051
LDL Cholesterol (mg/dL)	124.36 ± 31.89	117.22 ± 33.08	0.245
GGT (UI/L)	26.97 ± 16.26	22.74 ± 9.71	0.170
ALP (UI/L)	79.90 ± 23.04	86.14 ± 15.15	0.227
AST (UI/L)	21.88 ± 7.71	21.84 ± 7.71	0.981
AST (UI/L)	25.19 ± 14.45	21.22 ± 10.03	0.119
Creatinine (mg/dL)	0.78 ± 0.15	0.80 ± 0.16	0.624
TSH (mUI/L) FT3 (pg/mL) FT4 (ng/dL)	1.73 ± 0.98 3.39 ± 0.84 1.24 ± 0.41	2.14 ± 1.66 3.14 ± 0.56 1.35 ± 0.42	0.074 0.108 0.250

Abbreviations: HDL, high density lipoprotein; LDL, low density lipoprotein; GGT, gamma-glutamyl transpeptidase; ALP, alkaline phosphatase; AST, aspartate aminotransferase; ALT, alanine aminotransferase; TSH, thyroid-stimulating hormone; L-T4, Levothyroxine; *p* value, probability value.

**Table 3 medicina-58-00946-t003:** Step-wise, multivariable linear regression analysis- factors associated with Liver Stiffness.

C.I. 95%						
Dependent Variable						
*Liver Stiffness*						
	B	SE	β	*p*	LL	UL
L-T4	1.3	0.49	0.25	0.009	0.076	0.258

Abbreviations: L-T4, Levothyroxine; B, unstandardized coefficients; SE, standard error; ß, beta coefficient; *p*, probability value; CI, confidence interval; LL, lower limit; UL, upper limit. Excluded variables: BMI, age, glycaemia, triglycerides.

## Data Availability

The data presented in this study are available on request from thecorresponding author.

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
