# Peer review of "Liver Stiffness in Obese Hypothyroid Patients Taking Levothyroxine"

_medicina, 2022, doi:10.3390/medicina58070946_

Round 1

Reviewer 1 Report

The current manuscript is a well-written manuscript, with an excellent idea.

It opens the discussion for future studies - evolution, prediction of evolution. the question of the mechanism behind this difference remains open.

I would suggest looking un the cases with hypothyroidism = are there any differences among cases with know hypothyroidism onset  (Iatrogenic hypo) in which there is no long period with untreated hypothyroidism, and those with autoimmune disease, in which the length of untreated hypo is unknown.

Author Response

We thank reviewer for the precious suggestions, we modified the article according to suggestion as follow:

The current manuscript is a well-written manuscript, with an excellent idea.

It opens the discussion for future studies - evolution, prediction of evolution. the question of the mechanism behind this difference remains open.

I would suggest looking un the cases with hypothyroidism = are there any differences among cases with know hypothyroidism onset  (Iatrogenic hypo) in which there is no long period with untreated hypothyroidism, and those with autoimmune disease, in which the length of untreated hypo is unknown.

As suggested, we verified if there were any differences among different form of hypothyroidism and no difference between the group were found.

Reviewer 2 Report

Below are the reasons for not accepting the article as it was presented.

1.      Figure 1 is unnecessary once the same results are presented in Table 1.

2.      The authors do not explain the different obesity classes I, II and III shown in Figure 2.

3.      The data presented support that liver fibrosis is more prevalent in the group on Levo-T4 treatment; however, there are no data to support the findings on obesity. The non-obese group are not shown. If the authors want to obtain the relationship between liver fibrosis and obesity, at least they will need to have two groups (G1 - BMI >30 kg/m2 and G2 - BMI <30 kg/m2 ). Without presenting this data, the authors' conclusions cannot be supported.

Author Response

We thank reviewer for the precious suggestions, we modified the article according to suggestion as follow:

Below are the reasons for not accepting the article as it was presented.

  1. Figure 1 is unnecessary once the same results are presented in Table 1.

We removed fig.1  

  1. The authors do not explain the different obesity classesI, II and III shown in Figure 2.

We explained in the method section the different obesity classes 

  1. The data presented support that liver fibrosis is more prevalent in the group on Levo-T4 treatment; however, there are no data to support the findings on obesity. The non-obese group are not shown. If the authors want to obtain the relationship between liver fibrosis and obesity, at least they will need to have two groups(G1 - BMI >30 kg/m2 and G2 - BMI <30 kg/m2). Without presenting this data, the authors' conclusions cannot be supported.

We enclosed an analysis of 22 subjects recruited in previous trial on steatosis. All were in Levo-T4 treatment with similar age and female prevalence to the study group and with a BMI < 30 kg/m2. Unless they were all affected by steatosis (since it was the inclusion criterion) nobody had fibrosis.

We know that it is a not very rigorous method but the data confirmed the obesity role.  

Round 2

Reviewer 2 Report

The authors made improvements on the article as requested and I consider it suitable for publication in this journal.